# Comparison of the Meiofauna and Marine Nematode Communities before and after Removal of *Spartina alterniflora* in the Mangrove Wetland of Quanzhou Bay, Fujian Province

**Ming-Cheng Hu** [1], **Yu-Qing Guo** [1,2,*], **Yi-Jia Shih** [1,2], **Kai Liu** [1,2], **Chun-Xue Li** [1,2], **Fen-Fen Ji** [1,2] **and Ta-Jen Chu** [1,2]

1   Fisheries College, Jimei University, Xiamen 361021, China; humingcheng19098@163.com (M.-C.H.); eja0313@gmail.com (Y.-J.S.); liukai1218@jmu.edu.cn (K.L.); lichunxue1122@163.com (C.-X.L.); ffji1213266226@163.com (F-F.J.); chutajen@gmail.com (T.-J.C.)
2   Fujian Provincial Key Laboratory of Marine Fishery Resources and Eco-Environment, Jimei University, Xiamen 361021, China
*   Correspondence: guoyuqing@jmu.edu.cn

**Abstract:** The invasion of *Spartina alterniflora* is one of the main threats faced by mangrove wetlands in Quanzhou Bay, Fujian. To effectively manage *S. alterniflora*, mangrove plants (*Kandelia obovata, Aegiceras corniculatum, Bruguiera gymnorhiza, Rhizophora stylosa, and Avicennia marina)* were used to replace it in 2022 to restore the wetland ecosystem. Samples of meiofauna and marine nematodes were collected four times, including in September 2022 (before the removal of *S. alterniflora*), October 2022 (after removal), and December 2022 and March 2023 (after planting mangrove plants). This paper investigates changes in the composition, abundance, and biomass of meiofauna in different time periods, with a specific focus on comparing the community structure and biodiversity indices of marine nematodes in *S. alterniflora* and mangrove habitats. The results indicate that among the five meiofauna groups identified, marine nematodes account for 92.91%, 91.91%, 92.03%, and 85.92% of the total meiofauna abundance in the respective study periods. In the *S. alterniflora* habitat in September, marine nematodes were identified belonging to 12 families and 20 genera, of which 6 genera were dominant (percentage ≥ 5%). They were *Ptycholaimellus, Parodontophora, Terschellingia, Halichoanolaimus, Metachromadora*, and *Parasphaerolaimus*. In the mangrove habitat in December, marine nematodes were identified belonging to 15 families and 23 genera, with 6 genera being dominant, namely *Daptonema, Admirandus, Parodontophora, Ptycholaimellus, Terschellingia*, and *Anoplostoma*. Comparing the marine nematode communities in the two habitats, the mangrove habitat exhibits higher diversity than the *S. alterniflora* habitat. There was a change in the dominant genera, and their dominance has decreased. The dominant genera of marine nematodes found in both habitats are common and widely distributed groups. The changes in the abundance of meiofauna and the community structure of marine nematodes following the restoration of the *S. alterniflora* habitat by planting mangroves provide valuable insights for ecological monitoring after restoration measures in estuarine wetland conservation areas.

**Keywords:** mangrove; *Spartina alterniflora*; meiofauna; marine nematode

## 1. Introduction

Mangrove is the collective term for trees and shrub forests that grow in tropical and subtropical intertidal zones and tidal flats, and are flooded by periodic seawater. They serve important ecological functions, including protecting coasts, maintaining biodiversity, purifying seawater, sequestering, and storing carbon, and is one of the most productive marine ecosystems, with irreplaceable roles of terrestrial forests [1]. However, mangroves are also one of the ecosystems currently facing severe threats, especially in the Asian and Pacific regions. Aquaculture, over-exploitation of wood resources, and the invasion of

non-native species have led to the extensive degradation of mangroves, with approximately 70% of mangrove areas being destroyed [2–4]. *S. alterniflora* is a perennial herb native to the east coast of North America and the Gulf of Mexico. Compared with other mudflat plants, it is more tolerant to abiotic environmental stresses such as temperature salinity [5,6]. Due to *S. alterniflora*'s strong adaptability and diffusion ability, it spread rapidly from the introduction area and occupied the vast coastal beaches of China. At present, its distribution extends from Liaoning province in the north to Guangdong province in the south, high-risk areas for the invasion of *S. alterniflora* in mangroves are concentrated in Zhejiang and Fujian province, and it has become one of the main invasive plants in China's coastal wetlands [7–10]. In 2003, it was included in the list of 16 invasive alien species in China, and its invasion status and ecological effects on coastal zones have become a research hotspot. To effectively control the spread of *S. alterniflora* and ensure the ecological security of coastal wetlands, China initiated a special management action plan for the prevention and control of *S. alterniflora* [11,12].

Meiofauna is an indispensable component of coastal ecosystems. Free-living marine nematodes (hereinafter referred to as marine nematodes) are the dominant group among meiofauna, often accounting for more than 90 percent of its abundance [13–15]. In the mangrove benthic micro-food web, marine nematodes serve as a vital link in the nutritional chain between primary producers and large benthic animals, playing a significant role in material cycling and energy transfer [16–18]. Furthermore, their characteristics, such as high species diversity, wide distribution, and short life cycles, make them an important tool for detecting environmental changes and assessing ecosystem health [19,20].

The Quanzhou Bay Estuary Wetland Provincial Nature Reserve was established in 2003. This wetland is located at the mouth of the Jinjiang River and Luoyang River, with a total area of 7065.31 hectares. Among them, mangroves cover an area of 305.89 hectares. It is the largest mangrove forest with an artificial restoration area, and it is also the northern boundary of the natural distribution of *Avicennia marina* and *Aegiceras corniculatum* in China [21]. The introduction of *S. alterniflora* began in 1982 for the purpose of wave reduction and beach protection. In 2005, *S. alterniflora* had already spread throughout the entire Quanzhou Bay estuarine wetland, covering an area of 563 hectares and expanding outward each year. It became the most significant invasive plant in Quanzhou Bay and its adjacent coastal areas, posing a severe threat to the coastal mudflat ecosystems. The invasion of *S. alterniflora* not only squeezes out space for other plants but also disrupts the habitat for meiofauna, fish, and bird species. This invasion alters the structure of coastal mudflat ecosystems, leading to degradation, reduced biodiversity, and posing a significant threat to the ecological safety of China's coastal wetland ecosystems. [22]. The Quanzhou Bay Estuary Wetland Provincial Nature Reserve, where this study was conducted, is the experimental site for China's special management action to remove *S. alterniflora*. To prevent harm to the ecosystem, the government began to use physical methods such as felling in October 2022 to clear the *S. alterniflora* and replace it with mangroves.

There are many global studies on the abundance of meiofauna in mangrove wetlands, the structure of marine nematode communities, and the classification of marine nematodes [23–26]. There are also reports on research on marine nematode communities after the invasion of *S. alterniflora* in coastal wetlands. Fu et al. examined changes in the diversity of marine nematode communities in native mangrove wetland and the *S. alterniflora*-invaded area in the Zhangjiang River Estuary of Fujian, China [27]. Cao et al. compared soil nematode communities (mainly marine nematodes) in saltmarsh wetlands with *S. alterniflora*, reed, and Scirpus mariqueter communities in the Yangtze River Estuary, China [28]. Chen et al. conducted research comparing the genetic lineage structure, species diversity, and functional group composition of benthic nematodes in native plant communities and *S. alterniflora*-invaded communities in coastal saltmarsh wetlands and mangrove wetlands at five different latitudes in China [29]. However, there is a lack of research on the changes in the community structure of meiofauna and marine nematodes when coastal wetland vegetation is replaced by mangroves after the removal of *S. alterniflora*. This paper explores

the changes in the composition, abundance, and biomass of meiofauna, the community structure of marine nematodes, and their biodiversity in the Quanzhou Bay mangrove wetlands of Fujian, China, before and after the removal of *S. alterniflora*. The aim is to investigate the differences in the abundance of meiofauna and the community structure of marine nematodes in the early stages of the replacement of *S. alterniflora* by mangroves in estuarine wetlands. This research focuses on changes in the diversity of marine nematode communities and the dominant genera and aims to provide important reference data for understanding ecological environmental monitoring after wetland conservation and restoration measures in estuarine areas.

## 2. Materials and Methods

### 2.1. Study Area and Sampling Stations

This study involved a total of four samplings: before the removal of *S. alterniflora* (September 2022), after the removal of *S. alterniflora* (October 2022), and after planting of mangrove plants (December 2022 and March 2023). In the clearance area of *S. alterniflora*, five stations A, B, C, D, and E were arranged according to the different adaptability of various species to tidal environments, due to the planting of different mangrove species, while a control station CS was set up. The latitude and longitude of the stations are shown in Table 1, and the location map of the sampling stations is shown in Figure 1. Station A was planted with *Kandelia obovata*, *Aegiceras corniculatum*, *Rhizophora stylosa*, and *Avicennia marina* blocks, station B was planted with *K. obovata*, *A. corniculatum*, and *Bruguiera gymnorhiza* blocks, station C was planted with *K. obovata*, *A. corniculatum*, *B. gymnorhiza*, and *A. marina* blocks, stations D and E were only planted with *A. marina* blocks, and the control group station was the beach block without mangrove planting. Stations A and B are located at high tide levels, stations C and CS are at mid-tide level, and stations D and E are at low tide levels.

**Table 1.** Latitude and longitude of sampling areas in mangrove wetland of Quanzhou Bay.

| Sampling Station | Latitude | Longitude |
|---|---|---|
| A | 24.9213 | 118.6686 |
| B | 24.9222 | 118.6699 |
| C | 24.9210 | 118.6696 |
| D | 24.9184 | 118.6702 |
| E | 24.9177 | 118.6703 |
| CS | 24.9211 | 118.6687 |

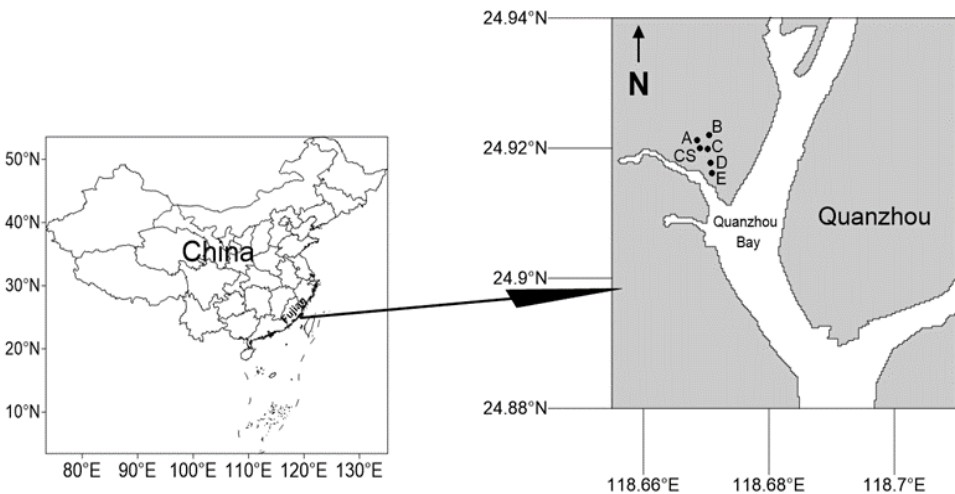

**Figure 1.** Sampling area map in mangrove wetland of Quanzhou Bay.

Two duplicates were set for each station. Locations with consistent sediment types and no disturbance were selected for sediment sample collection. A sampling tube with an inner diameter of 2.9 cm was used, and three core samples were collected at each station at a depth of 5 cm. After collecting the samples, they were fixed with DESS and stored at room temperature before being taken back to the laboratory for sample sorting.

### 2.2. Sample Handling and Sorting

The indoor sorting of meiofauna and mounted methods of marine nematodes have been described in the literature [30,31]. Each sediment sample was placed on a separation device consisting of a 500 μm upper screen and a 42 μm screen below. It was then slowly washed with filtered tap water until most of the clay and silt were removed. The retained sample was rinsed in a sieve with Ludox-TM silicone solution with a density of 1.15 g/mL into a centrifuge tube. The sample was then centrifuged twice at a speed of 4000 r/min for 10 min each time. The meiofauna extracted from the supernatant were transferred to Petri dishes with equal-width parallel lines. Under a dissecting microscope (Nikon SMZ800, Tokyo, Japan), the meiofauna were classified and counted by group. The marine nematodes were picked out into a special glass container containing a mixture of alcohol, glycerol, and water (V alcohol:V glycerol:V water = 1:1:18). After placing the glass container in a drying oven for one week, the marine nematodes were picked out onto a cover glass slide with an appropriate amount of mixture. Identification was conducted under a differential interference microscope (Nikon ECLI-PSE-80i).

### 2.3. Data Processing and Analysis

The data were processed and analyzed using the software programs Excel 2016, SPSS 22, and Primer 6.0. The data were analyzed in Primer 6.0 with DIVERSE: the Shannon–Wiener index (H′), Margalef's species richness index (D), Pielou's evenness index (J′), and the dominance index ($1 - \lambda$). The classification of meiofauna was mainly conducted with reference to the *Introduction to the Study of Meiofauna* [32]. The taxonomic identification of marine nematodes was mainly based on *Freeliving Marine Nematodes: Part III Monhysterida* [33].

The estimation of biomass was based on the average dry mass exchange algorithm of various groups [34]. The empirical coefficients used for each group were: marine nematode 0.826 μg [35], copepoda 1.86 μg [36], polychaeta 14 μg, oligochaeta 13.98 μg, and other unidentified taxa 3.5 μg [37].

## 3. Results

### 3.1. Composition, Abundance, and Biomass

Meiofauna were identified, including marine nematodes, copepoda, polychaeta, oligochaete, and turbellaria. Among them, polychaeta only appeared in September 2022 and March 2023. Marine nematodes were the absolutely dominant group in all four months, accounting for 92.91%, 91.91%, 92.03%, and 85.92% of the total number of meiofauna. The composition, abundance, and biomass of meiofauna are shown in Tables 2 and 3.

**Table 2.** Meiofauna group abundance in Quanzhou Bay mangrove wetland (ind./10 cm$^2$).

| Date | Group | Station A | Station B | Station C | Station D | Station E | Station CS | Mean |
|---|---|---|---|---|---|---|---|---|
| 2022.09 | Nematoda | 127.17 ± 155.58 | 179.91 ± 245.15 | 130.96 ± 102.41 | 327.52 ± 329.01 | 233.15 ± 107.05 | 78.98 ± 18.91 | 179.61 ± 159.69 |
| | Copepoda | 0.5 ± 0 | 11.35 ± 15.34 | 12.87 ± 16.77 | 0 ± 0 | 30.03 ± 2.5 | 30.28 ± 27.83 | 14.17 ± 10.41 |
| | Oligochaeta | 0.5 ± 0.71 | 0.5 ± 0.71 | 1.51 ± 1.43 | 0 ± 0 | 0.76 ± 1.07 | 0.25 ± 0.36 | 0.59 ± 0.71 |
| | Polychaeta | 0 ± 0 | 0 ± 0 | 1.51 ± 1.43 | 0 ± 0 | 0 ± 0 | 0 ± 0 | 0.25 ± 0.24 |
| | Turbellaria | 1.01 ± 0 | 0 ± 0 | 0 ± 0 | 0 ± 0 | 0 ± 0 | 0 ± 0 | 0.17 ± 0 |
| | Total | 129.19 ± 156.3 | 191.77 ± 259.78 | 146.85 ± 122.04 | 327.52 ± 329.01 | 263.93 ± 103.48 | 109.51 ± 46.39 | 194.8 ± 169.5 |
| 2022.10 | Nematoda | 14.89 ± 10.35 | 17.92 ± 12.49 | 31.79 ± 17.13 | 126.92 ± 57.45 | 97.9 ± 44.25 | 3.03 ± 2.14 | 48.74 ± 23.97 |
| | Turbellaria | 0.76 ± 1.07 | 1.01 ± 0.71 | 0.25 ± 0.36 | 9.08 ± 12.85 | 8.83 ± 3.93 | 0.5 ± 0.71 | 3.41 ± 3.27 |
| | Copepoda | 0 ± 0 | 0.76 ± 0.36 | 0 ± 0 | 0.25 ± 0.36 | 0 ± 0 | 4.04 ± 5.71 | 0.84 ± 1.07 |
| | Oligochaeta | 0 ± 0 | 0 ± 0 | 0 ± 0 | 0 ± 0 | 0.25 ± 0.36 | 0 ± 0 | 0.04 ± 0.06 |
| | Total | 15.64 ± 11.42 | 19.68 ± 11.42 | 32.05 ± 17.49 | 136.26 ± 70.66 | 106.99 ± 47.82 | 7.57 ± 4.28 | 53.03 ± 27.18 |
| 2022.12 | Nematoda | 118.59 ± 5 | 167.54 ± 42.82 | 59.55 ± 6.42 | 50.47 ± 28.55 | 110.27 ± 119.54 | 67.88 ± 19.63 | 95.72 ± 36.99 |
| | Turbellaria | 4.54 ± 5 | 0 ± 0 | 1.26 ± 1.78 | 0 ± 0 | 0 ± 0 | 0.5 ± 0.71 | 1.05 ± 1.25 |
| | Copepoda | 1.01 ± 0.71 | 4.79 ± 2.5 | 17.41 ± 22.48 | 0.25 ± 0.36 | 0.25 ± 0.36 | 0.25 ± 0.36 | 4 ± 4.46 |
| | Oligochaeta | 7.07 ± 4.28 | 1.77 ± 0.36 | 1.01 ± 0 | 4.29 ± 6.07 | 3.53 ± 5 | 1.77 ± 1.07 | 3.24 ± 2.8 |
| | Total | 131.21 ± 3.57 | 174.11 ± 40.68 | 79.23 ± 27.12 | 55.01 ± 34.97 | 114.05 ± 124.18 | 70.40 ± 21.77 | 104.00 ± 42.05 |
| 2023.03 | Nematoda | 357.8 ± 189.84 | 679.77 ± 220.53 | 811.48 ± 0.71 | 695.16 ± 335.79 | 1629.52 ± 479.6 | 1339.1 ± 246.58 | 918.8 ± 245.51 |
| | Turbellaria | 1.77 ± 0.36 | 0 ± 0 | 3.53 ± 4.28 | 0.25 ± 0.36 | 2.02 ± 2.85 | 6.06 ± 2.85 | 2.27 ± 1.78 |
| | Copepoda | 74.18 ± 24.98 | 86.8 ± 27.12 | 33.56 ± 1.78 | 183.19 ± 122.75 | 219.52 ± 13.56 | 275.79 ± 23.19 | 145.51 ± 35.57 |
| | Oligochaeta | 2.52 ± 1.43 | 0 ± 0 | 0.76 ± 0.36 | 0 ± 0 | 0 ± 0 | 13.37 ± 3.93 | 2.78 ± 0.95 |
| | Polychaeta | 0 ± 0 | 0 ± 0 | 0 ± 0 | 0.25 ± 0.36 | 0 ± 0 | 0 ± 0 | 0.04 ± 0.06 |
| | Total | 436.27 ± 216.6 | 766.57 ± 247.65 | 849.33 ± 5 | 878.85 ± 459.26 | 1851.07 ± 463.18 | 1634.32 ± 230.16 | 1069.40 ± 270.31 |

Table 3. Meiofauna group biomass in Quanzhou Bay mangrove wetland ($\mu g/10\ cm^2$).

| Date | Group | Station A | Station B | Station C | Station D | Station E | Station CS | Mean |
|------|-------|-----------|-----------|-----------|-----------|-----------|------------|------|
| 2022.09 | Nematoda | 101.74 ± 124.47 | 143.93 ± 196.12 | 104.77 ± 81.93 | 262.02 ± 263.21 | 186.52 ± 85.64 | 63.18 ± 15.13 | 143.69 ± 127.75 |
| | Copepoda | 0.94 ± 0 | 21.12 ± 28.54 | 23.94 ± 31.2 | 0 ± 0 | 55.85 ± 4.65 | 56.32 ± 51.77 | 26.36 ± 19.36 |
| | Oligochaeta | 7.06 ± 9.98 | 7.06 ± 9.98 | 21.17 ± 19.95 | 0 ± 0 | 10.58 ± 14.97 | 3.53 ± 4.99 | 8.23 ± 9.98 |
| | Polychaeta | 0 ± 0 | 0 ± 0 | 21.2 ± 19.98 | 0 ± 0 | 0 ± 0 | 0 ± 0 | 3.53 ± 3.33 |
| | Turbellaria | 3.53 ± 0 | 0 ± 0 | 0 ± 0 | 0 ± 0 | 0 ± 0 | 0 ± 0 | 0.59 ± 0 |
| | Total | 113.26 ± 134.44 | 172.1 ± 214.68 | 171.06 ± 153.06 | 262.02 ± 263.21 | 252.95 ± 66.03 | 123.03 ± 61.91 | 182.4 ± 148.89 |
| 2022.10 | Nematoda | 11.91 ± 8.28 | 14.33 ± 9.99 | 25.43 ± 13.7 | 101.54 ± 45.96 | 78.32 ± 35.4 | 2.42 ± 1.71 | 38.99 ± 19.17 |
| | Turbellaria | 2.65 ± 3.75 | 3.53 ± 2.5 | 0.88 ± 1.25 | 31.79 ± 44.96 | 30.91 ± 13.74 | 1.77 ± 2.5 | 11.92 ± 11.45 |
| | Copepoda | 0 ± 0 | 1.41 ± 0.66 | 0 ± 0 | 0.47 ± 0.66 | 0 ± 0 | 7.51 ± 10.62 | 1.56 ± 1.99 |
| | Oligochaeta | 0 ± 0 | 0 ± 0 | 0 ± 0 | 0 ± 0 | 3.53 ± 4.99 | 0 ± 0 | 0.59 ± 0.83 |
| | Total | 14.56 ± 12.03 | 19.27 ± 6.83 | 26.32 ± 14.95 | 133.8 ± 91.59 | 112.76 ± 44.15 | 11.7 ± 11.4 | 53.07 ± 30.16 |
| 2022.12 | Nematoda | 94.87 ± 4 | 134.04 ± 34.26 | 47.64 ± 5.14 | 40.37 ± 22.84 | 88.21 ± 95.63 | 54.3 ± 15.7 | 76.57 ± 29.59 |
| | Turbellaria | 15.9 ± 17.49 | 0 ± 0 | 4.42 ± 6.24 | 0 ± 0 | 0 ± 0 | 1.77 ± 2.5 | 3.68 ± 4.37 |
| | Copepoda | 1.88 ± 1.33 | 8.92 ± 4.65 | 32.38 ± 41.81 | 0.47 ± 0.66 | 0.47 ± 0.66 | 0.47 ± 0.66 | 7.43 ± 8.3 |
| | Oligochaeta | 98.77 ± 59.86 | 24.69 ± 4.99 | 14.11 ± 0 | 59.97 ± 84.81 | 49.39 ± 69.84 | 24.69 ± 14.97 | 45.27 ± 39.08 |
| | Total | 211.42 ± 45.05 | 167.65 ± 34.6 | 98.55 ± 40.71 | 100.81 ± 108.31 | 138.07 ± 164.81 | 81.23 ± 33.83 | 132.59 ± 71.22 |
| 2023.03 | Nematoda | 286.24 ± 151.87 | 543.81 ± 176.42 | 649.19 ± 0.57 | 556.13 ± 268.63 | 1303.62 ± 383.68 | 1071.28 ± 197.26 | 735.04 ± 196.41 |
| | Turbellaria | 6.18 ± 1.25 | 0 ± 0 | 12.36 ± 14.99 | 0.88 ± 1.25 | 7.07 ± 9.99 | 21.2 ± 9.99 | 7.95 ± 6.24 |
| | Copepoda | 137.98 ± 46.46 | 161.45 ± 50.44 | 62.42 ± 3.32 | 340.73 ± 228.32 | 408.31 ± 25.22 | 512.97 ± 43.14 | 270.65 ± 66.15 |
| | Oligochaeta | 35.28 ± 19.95 | 0 ± 0 | 10.58 ± 4.99 | 0 ± 0 | 0 ± 0 | 186.96 ± 54.88 | 38.8 ± 13.3 |
| | Polychaeta | 0 ± 0 | 0 ± 0 | 0 ± 0 | 3.53 ± 5 | 0 ± 0 | 0 ± 0 | 0.59 ± 0.83 |
| | Total | 465.68 ± 219.54 | 705.26 ± 226.87 | 734.55 ± 12.75 | 901.27 ± 503.2 | 1719 ± 348.46 | 1792.41 ± 218.99 | 1053.03 ± 254.97 |

The total average abundance of meiofauna ranged from 53.03 ± 27.18 to 1069.40 ± 270.31 ind./10 cm$^2$. The highest abundance occurred in March 2023. The lowest abundance occurred in October 2022. The abundance trend of meiofauna across the different months was March 2023 > September 2022 > December 2022 > October 2022. The results of a one-way ANOVA showed that there was a significant difference in the abundance of meiofauna in the different periods (F = 30.530, df = 3, *p* < 0.01). Then, after multiple comparisons (LSD), there were significant differences between March 2023 and the other periods September 2022, October 2022, and December 2022 (*p* < 0.01). Furthermore, there was no significant difference in meiofauna abundance among the stations (F = 0.614, df = 5, *p* > 0.05).

The total average biomass of the meiofauna ranged from 53.07 ± 30.16 to 1053.03 ± 254.97 µg/10 cm$^2$. The highest biomass occurred in March 2023. The lowest biomass occurred in October 2022. The trend of meiofauna biomass across the different months was consistent with the trend of meiofauna change. The results of a one-way ANOVA showed that there was a significant difference in the biomass of meiofauna across the different periods (F = 28.937, df = 3, *p* < 0.01). Then, after multiple comparisons (LSD), there were significant differences between March 2023 and the other periods September 2022, October 2022, and December 2022 (*p* < 0.01). Furthermore, the biomass of meiofauna was not significantly different among the stations (F = 0.625, df = 5, *p* > 0.05).

Notably, this clearly shows that the abundance and biomass of meiofauna changed just after the *S. alterniflora* was removed.

### 3.2. Marine Nematode Abundance and Biomass

From September 2022 to March 2023, the average abundance of marine nematodes was 179.61 ± 159.69 ind./10 cm$^2$, 48.74 ± 23.97 ind./10 cm$^2$, 95.72 ± 36.99/10 cm$^2$, and 918.80 ± 245.51 ind./10 cm$^2$, respectively. The average biomass of marine nematodes was 143.69 ± 127.75 µg/10 cm$^2$, 38.99 ± 19.17 µg/10 cm$^2$, 76.57 ± 29.59 µg/10 cm$^2$, and 735.04 ± 196.41 µg/10 cm$^2$, respectively. The average abundance and the maximum average biomass occurred in March 2023, and the minimum values occurred in October 2022. The results of a one-way ANOVA showed that there was a significant difference in abundance and biomass between marine nematodes in each month (F = 28.650, df = 3, *p* < 0.01). Also, there was no significant difference in abundance or biomass between the marine nematodes at each station (F = 0.604, df = 5, *p* > 0.05).

### 3.3. Marine Nematode Dominant Genus

The marine nematode community structure was obtained using samples from September 2022 and December 2022. The genera and dominances are shown in Table 4. In September, a total of 12 families and 20 genera of marine nematodes were identified, including 6 dominant genera (quantity percentage ≥ 5%), namely *Ptycholaimellus*, *Parodontophora*, *Terschellingia*, *Halichoanolaimus*, *Metachromadora*, and *Parasphaerolaimus*, with dominances of 26.54%, 19.57%, 13.22%, 5.98%, 5.25%, and 5.07%, respectively, together accounting for 75.63% of the total abundance. In December, a total of 23 genera belonging to 15 families were identified, including 6 dominant genera, which were *Daptonema*, *Admirandus*, *Parodontophora*, *Ptycholaimellus*, *Terschellingia,* and *Anoplostoma*, with dominances of 15.95%, 12.36%, 11.37%, 9.77%, 9.17%, and 8.47%, respectively. In total, they accounted for 67.10% of the total abundance. There were three common dominant genera in the two months, namely *Ptycholaimellus*, *Parodontophora*, and *Terschellingia*, but their dominances changed from 26.54% to 9.77%, 19.57% to 11.37%, and 13.22% to 9.17%. The dominances of the common genera *Halichoanolaimus*, *Metachromadora*, and *Parasphaerolaimus* decreased from 5.98% to 1.4%, 5.25% to 3.39%, and 5.07% to 0.70%, respectively. The most dominant genus *Daptonema* in December did not appear in September, and the dominances of two common genera *Admirandus* and *Anoplostoma* gradually increased from September, from 1.00% to 12.36% and 3.80% to 8.47%, respectively, becoming the dominant genera.

**Table 4.** Dominant genera of marine nematodes in Quanzhou Bay mangrove wetland.

| Dominant Genera | Percentage of Dominances (%) | | | | | | | | | | | |
|---|---|---|---|---|---|---|---|---|---|---|---|---|
| | Station A | | Station B | | Station C | | Station D | | Station E | | Station CS | |
| | Sep. | Dec. | Sep. | Dec. | Sep. | Dec. | Sep. | Dec. | Sep. | Dec. | Sep. | Dec. |
| *Ptycholaimellus* | 2.00 | 7.76 | 6.43 | 22.31 | 53.30 | 22.37 | 34.22 | 1.30 | 11.83 | 1.13 | 53.37 | 4.55 |
| *Parodontophora* | 11.00 | 18.10 | 26.32 | 4.55 | 20.81 | 2.63 | 31.55 | 14.29 | 11.83 | 8.65 | 16.56 | 22.73 |
| *Terschellingia* | 49.00 | - | 9.94 | - | 2.54 | 1.32 | 5.35 | 35.06 | 6.45 | 9.02 | 2.45 | 36.36 |
| *Halichoanolaimus* | - | 2.59 | 25.15 | - | 5.08 | - | 2.14 | 6.49 | 3.23 | 1.13 | 1.84 | - |
| *Metachromadora* | 4.50 | 4.74 | 3.51 | - | 3.05 | - | - | 7.79 | 18.28 | 6.39 | 1.84 | - |
| *Parasphaerolaimus* | 1.00 | - | 5.26 | - | 0.51 | 1.32 | - | - | 20.97 | 2.26 | 3.07 | - |
| *Daptonema* | - | - | - | - | - | - | - | - | - | 56.39 | - | 10.00 |
| *Admirandus* | - | 3.02 | - | 39.26 | 0.51 | 28.95 | - | - | 4.30 | - | 1.23 | - |
| *Anoplostoma* | - | 14.66 | 2.34 | 18.60 | 7.11 | 3.95 | 2.67 | 3.90 | 0.54 | - | 11.04 | - |

Note: "-" indicate that the genera did not exist at the sampling station in the given month.

### 3.4. Marine Nematode Diversity and K-Dominance Analysis

The diversity analysis of marine nematodes at each station in September 2022 and December 2022 are shown in Figure 2 and Table 5. The Shannon–Wiener index (H') of marine nematodes ranges from 1.51 to 2.49. Margalef's species richness index (D) ranges from 1.91 to 3.12. Pielou's evenness index (J') is between 0.61 and 0.86, and the dominance index $(1 - \lambda)$ is between 0.66 and 0.90. The highest species number and Margalef's species richness index appeared at station B in September, while the highest Shannon–Wiener index, Pielou's evenness index, and dominance index all appeared at station E. The lowest species number and Margalef's species richness index appeared at station D in September, while the lowest Shannon–Wiener index, Pielou's evenness index, and dominance index all appeared at station C. The highest species number, Margalef's species richness index, Shannon–Wiener index, Pielou's evenness index, and dominance index in December all appeared at station A. The lowest species number appeared at stations D and CS. The lowest Margalef's species richness index appeared at station D. The lowest Shannon–Wiener index, Pielou's evenness index, and dominance index all appeared at station B. Overall, the dominance curve of station C in September was at the top and that of station A in December at the bottom. The K-dominance analysis results of the other stations are between station C and A. The dominance of the dominant genera of marine nematodes in September was higher than that in December, while species diversity was lower than that in December.

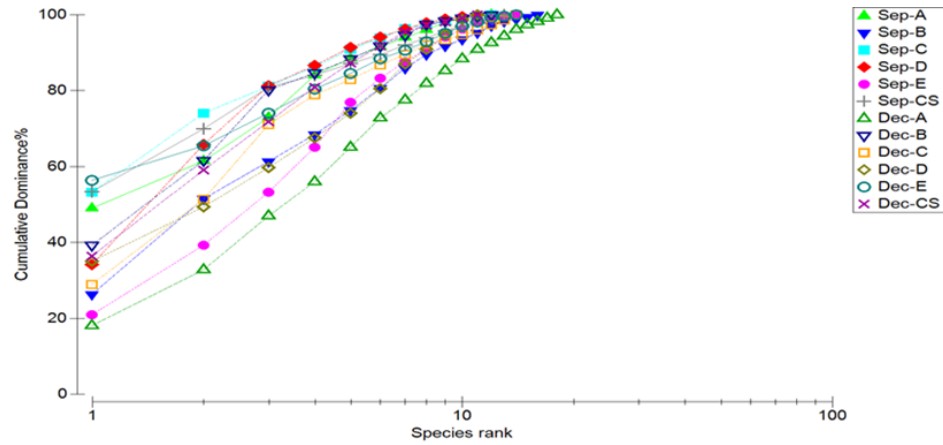

**Figure 2.** K-dominance curves of marine nematode abundances at different stations and in different months in Quanzhou Bay mangrove wetland.

**Table 5.** Biodiversity index of marine nematodes in Quanzhou Bay mangrove wetland.

| Station | Species Number (S) | Margalef's Species Richness Index (D) | Pielou's Evenness Index (J') | Shannon–Wiener Index (H') | Dominance Index (1 − λ) |
|---|---|---|---|---|---|
| Sep-A | 12 | 2.08 | 0.68 | 1.69 | 0.72 |
| Sep-B | 16 | 2.92 | 0.78 | 2.17 | 0.84 |
| Sep-C | 12 | 2.08 | 0.61 | 1.51 | 0.66 |
| Sep-D | 11 | 1.91 | 0.7 | 1.67 | 0.76 |
| Sep-E | 14 | 2.49 | 0.84 | 2.21 | 0.87 |
| Sep-CS | 13 | 2.36 | 0.63 | 1.61 | 0.68 |
| Dec-A | 18 | 3.12 | 0.86 | 2.49 | 0.9 |
| Dec-B | 12 | 2 | 0.69 | 1.72 | 0.76 |
| Dec-C | 14 | 3 | 0.76 | 2 | 0.83 |
| Dec-D | 11 | 2.3 | 0.85 | 2.05 | 0.83 |
| Dec-E | 14 | 2.32 | 0.63 | 1.66 | 0.66 |
| Dec-CS | 11 | 2.12 | 0.76 | 1.82 | 0.79 |

### 3.5. Cluster Analysis

The cluster analysis of the marine nematode communities is shown in Figure 3. The species similarity of marine nematodes at each station was between 26.39 and 81.67%. At the similarity level of 30.91%, the 12 stations studied in the two months were divided into three groups: Dec-B and Dec-C in a group; Sep-B, Sep-C, Sep-D, Sep-E, Sep-CS, and Dec-A in a group; and Sep-A, Dec-D, Dec-E, and Dec-CS in a group. The reasons for this result may be closely related to the season. Among them, the nematode community structure of Dec-B and Dec-C were quite different from that of the other stations, and the lowest similarity was 33.96%. Sep-C and Sep-CS had the most similar nematode community structures, with a similarity of 81.67%. From the identification of marine nematodes, Sep-C and Sep-CS stations had six common dominant genera.

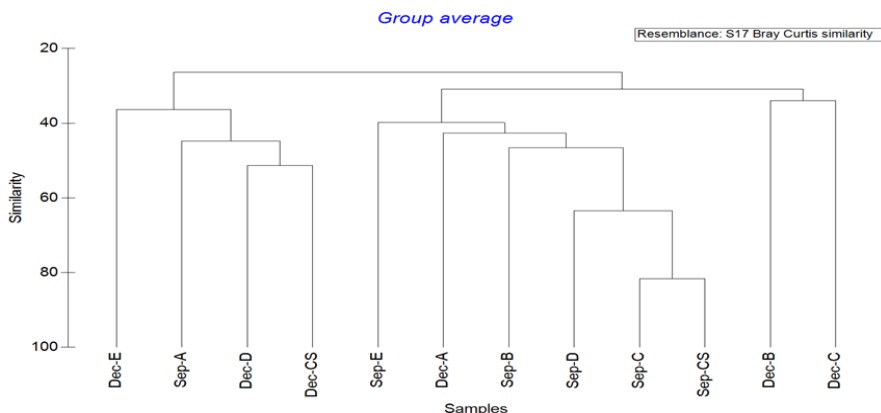

**Figure 3.** Cluster analysis map of marine nematode communities in Quanzhou Bay mangrove wetland.

## 4. Discussion

### 4.1. Composition, Abundance, and Biomass of Meiofauna before and after Removal of Spartina alterniflora

In this study, a total of five meiofauna were identified, namely marine nematodes, copepoda, turbellaria, polychaeta, and oligochaeta. Polychaeta only appeared at station C in September 2022 and station D in March 2023. Among them, marine nematodes are an absolutely dominant group, accounting for 85.92~92.91% of the meiofauna. Chang et al. identified a total of 11 groups of meiofauna in the mangrove forest of Luoyang Estuary, Fujian Province. Among them, marine nematodes accounted for 90.53~97.02%, and the proportion of marine nematodes was more consistent with this study, but there were more

groups [38]. The reason may be that there was no invasion of *S. alterniflora* and human interference in the Luoyang Estuary Mangrove Nature Reserve, Fujian.

The lowest values for both the total average abundance and the total average biomass of meiofauna occurred in October 2022. It may be that some meiofauna lost good habitat and food sources after *S. alterniflora* was removed, or that the abundance and biomass declined due to anthropogenic disturbance and habitat changes [39,40]. The highest values occurred in March 2023. This may be due to the gradual stabilization of sedimentary habitats and an increase in organic matter content in sediments after mangrove cultivation. The meiofauna have gained good shelter and abundant food sources, leading to an increase in abundance and biomass [41].

Due to the rich presence of tannins and organic matter in mangrove ecosystems, Sahoo et al. [42] suggested that meiofauna in mangrove areas exhibit distinct differences from non-mangrove regions. Research by Gee et al. [43] indicated that the community structure of marine nematodes is associated with the sediment formed by different mangrove plants. In a study conducted in the Xinying Port mangrove wetlands of Hainan, it was found that the abundance and community composition of marine nematodes differed between two distinct habitats: the *Bruguiera sexangular* community and the mixed community of *B. gymnorhiza* and *A. marina* [44]. Although different mangrove species were planted at different stations in this study, the relatively short duration of mangrove growth had not yet revealed any differences in marine nematode communities among the different mangrove plants.

*4.2. A Comparison of Marine Nematode Diversity and Dominant Genera between S. alterniflora and Mangrove Habitats*

Fu et al. [27] found that the number of species of marine nematodes in the *S. alterniflora* community was the lowest compared with the native mangrove community in the Zhangjiang River Estuary mangrove wetland. At the same time, the species richness index, average diversity index, and maturity index were also the lowest. This indicates that the invasion of *S. alterniflora* has significantly disturbed mangrove ecosystems. However, the differences in biodiversity indices in this study were not significant. This may be due to the short planting time of mangroves and the little change in marine nematode communities.

In the *S. alterniflora* habitat of this study in September, the dominant genera of marine nematodes were *Ptycholaimellus*, *Parodontophora*, *Terschellingia*, *Halichoanolaimus*, *Metachromadora,* and *Parasphaerolaimus*. Fu et al. [27], in their study of the *S. alterniflora*-invaded area in the Zhangjiang River Estuary mangrove wetlands, found the dominant genera to be *Parodontophora*, *Metachromadora*, *Sphaerolaimus*, *Spliophorella*, *Daptonema*, *Terschellingia*, and *Sabatieria*. This study shares common dominant genera with *Parodontophora*, *Terschellingia*, and *Metachromadora*. Chen et al. [29], in their study of the *S. alterniflora*-invaded area in the Yangtze River Estuary saltmarsh wetlands, reported the dominant genera as *Ethmolaimus*, *Terschellingia*, *Haliplectus,* and *Parodontophora*, sharing *Terschellingia* and *Parodontophora* as common dominant genera with this study. Cao [28], in research conducted in the Zhangjiang River Estuary mangrove conservation area in Fujian Province during the summer, noted the dominant genera as *Sabatieria*, *Ptycholaimellus*, *Onyx*, *Parodontophora,* and *Hypodontolaimus*, sharing *Ptycholaimellus* and *Parodontophora* as common dominant genera with this study. In the *S. alterniflora* area in Ximen Island, Leqing Bay, Wenzhou, Zhejiang Province, China, the dominant genera of marine nematodes included *Onyx*, *Anoplostoma*, *Terschellingia*, *Ptycholaimellus,* and *Sabatieria*, with the common dominant genera of *Terschellingia* and *Ptycholaimellus* shared with this study.

Among mangrove wetland sediments around the world, *Daptonema*, *Theristus*, *Dorylaimopsis*, *Hopperia*, *Ptycholaimellus*, *Terschellingia*, *Sabatieria*, *Anoplostoma*, and *Parodontophora* are common marine nematode dominant genera [45]. In the mangrove habitat, this study identified six dominant genera in December, which included *Daptonema*, *Admirandus*, *Parodontophora*, *Ptycholaimellus*, *Terschellingia,* and *Anoplostoma*. Among these, *Daptonema*, *Parodontophora*, *Ptycholaimellus*, *Terschellingia,* and *Anoplostoma* are the five com-

mon dominant genera of marine nematodes. In the research of Nicholas et al. [46] on the mangrove wetlands of southeastern Australia, a total of nine dominant genera were identified, including *Ptycholaimellus*, *Desmodora*, *Microlaimus*, *Sphaerolaimus*, *Terschellingia*, *Parodontophora*, *Onyx*, *Daptonema,* and *Sabatieria*. This study shares four common dominant genera: *Daptonema*, *Parodontophora*, *Ptycholaimellus,* and *Terschellingia*. In the study of Mokievsky et al. [47] in Nha Trang Bay, Vietnam, a total of nine dominant genera were identified, including *Chromadorella*, *Ptycholaimellus*, *Admirandus*, *Theristus*, *Sabatieria*, *Haliplectus*, *Anoplostoma,* and *Litinium*. This research shares three common dominant genera with that study: *Ptycholaimellus*, *Admirandus,* and *Anoplostoma.*

In research conducted in five mangrove wetlands in Fujian Province, a total of 12 dominant genera were identified, including *Sabatieria*, *Ptycholaimellus*, *Parasphaerolaimus*, *Terschellingia*, *Daptonema*, *Viscosia*, *Dichromadora*, *Anoplostoma*, *Spilophorella*, *Trissonchulus*, *Hopperia,* and *Sphaeroalaimus*. This study shares four common dominant genera with that research: *Ptycholaimellus*, *Terschellingia*, *Daptonema,* and *Anoplostoma* [48]. Chang [45] identified a total of eight dominant genera in the Luoyang Estuary mangrove wetland, namely, *Sabatieria*, *Parasphaerolaimus*, *Viscosia*, *Hopperia*, *Doptonema*, *Terschellingia*, *Ptycholaimellus,* and *Trissonchulus*. This study shares three common dominant genera: *Daptonema*, *Terschellingia,* and *Ptycholaimellus*. In Xiao et al.'s [49] study of the Kinmen mangrove wetlands, a total of eight dominant genera were identified, which include *Ptycholaimellus*, *Viscosia*, *Daptonema*, *Dorylaimopsis*, *Parasphaerolaimus*, *Anoplostoma*, *Metachromadora,* and *Spilophorella*. This research shares three common dominant genera with that study: *Ptycholaimellus*, *Daptonema,* and *Anoplostoma*. These results demonstrate that mangrove wetland habitats worldwide exhibit a degree of similarity in the community structure of marine nematodes. They also indicate that, after planting mangrove plants for three months, the dominant genera that appear in the marine nematode community are widely distributed taxa on a global scale.

This study identified common dominant genera, including *Ptycholaimellus*, *Parodontophora,* and *Terschellingia*, in both the *S. alterniflora* habitat and the mangrove habitat. In Fu et al.'s research, it was found that in different habitats with *S. alterniflora* and different mangrove plants, common dominant genera included *Daptonema*, *Spilophorella*, *Terschellingia,* and *Parodontophora* in *S. alterniflora* and *K. obovata* habitats, *Sabatieria*, *Spilophorella,* and *Terschellingia* in *S. alterniflora* and *B. gymnorhiza* habitats, and *Spilophorella*, *Parodontophora*, *Daptonema*, *Metachromadora,* and *Terschellingia* in *S. alterniflora* and *A. marina* habitats. *Hypodontolaimus* was a dominant genus in both the mangrove and *S. alterniflora*-invaded areas in the Zhangjiang Estuary mangrove conservation area in Fujian, China [27]. *Terschellingia* and *Ptycholaimellus* were dominant genera in both the mangrove and *S. alterniflora*-invaded areas in Ximen Island, Leqing Bay, Wenzhou, Zhejiang Province, China [28]. These studies suggest that there is a degree of similarity in the dominant genera of marine nematode communities between *S. alterniflora* and different mangrove plant habitats.

In this study, the dominances of the *Terschellingia* genus were 13.22% and 9.17%, and the dominances of the *Parodontophora* genus were 19.57% and 11.37%, respectively. *Daptonema* did not appear in the September samples but became the dominant genus in December with a dominance of 15.95%. Previous research has indicated that the dominance of *Daptonema*, *Terschellingia,* and *Parodontophora* is often used as a crucial indicator for assessing ecological quality, with a relative abundance of >10% indicating poor ecological conditions [50–52]. Similar results were also found in Hua et al.'s study of Shenzhen Futian Mangrove Reserve [53]. Therefore, it can be inferred that the Quanzhou Bay mangrove wetlands may be facing significant organic matter pollution.

In the southern Fujian region, there is ecological niche overlap between *S. alterniflora* and mangroves, indicating a competitive relationship in terms of spatial distribution. Mangrove plants can occupy the living space of *S. alterniflora*, eventually leading to a biological replacement outcome [54]. Currently, no natural limiting factors have been identified to effectively control the spread of *S. alterniflora*, so the management of this species appears to require human intervention [55]. The preliminary findings of this study

suggest that a bio-remediation approach involving initial cutting and subsequent planting of mangrove vegetation can enhance the abundance of meiofauna and the biodiversity of marine nematodes in the sediment environment. This method holds the potential to gradually restore the wetland ecosystem functions in estuarine conservation areas.

## 5. Conclusions

This study identified five groups of meiofauna, with marine nematodes being the overwhelmingly dominant group before and after the removal of *S. alterniflora*. There were significant differences in the abundance and biomass of meiofauna and marine nematodes among the different sampling time periods, while no significant differences were observed among the different sampling stations. Comparing *S. alterniflora* and mangrove habitats, the diversity of marine nematodes was higher in the mangrove habitat than in the *S. alterniflora* habitat, with a change in dominant genera and reduced dominance. Both *S. alterniflora* and mangrove habitats shared a certain number of common dominant genera in their marine nematode communities. Although this study introduced different species of mangrove plants at various locations, the short planting time makes it difficult to discern differences in the marine nematode community structures among these mangrove species at present. Furthermore, it should be noted that this study presents preliminary results of the project, focusing on the diversity and dominant genera of early-stage marine nematode communities during restoration. It cannot provide explanations or discussions regarding the correlation between sediment environmental factors and biological factors. In the later stages of the project, with the accumulation of a significant amount of biological and sediment environmental data, the focus will be on studying the correlation between them.

**Author Contributions:** Conceptualization, Y.-Q.G.; methodology, Y.-Q.G. and T.-J.C.; software, Y.-Q.G., M.-C.H. and Y.-J.S.; validation, Y.-Q.G. and Y.-J.S.; formal analysis, Y.-Q.G., Y.-J.S. and T.-J.C.; investigation, M.-C.H., Y.-J.S., K.L., C.-X.L. and F.-F.J.; resources, Y.-Q.G. and Y.-J.S.; data curation, M.-C.H. and Y.-Q.G.; writing—original draft preparation, M.-C.H.; writing—review and editing, Y.-Q.G. and T.-J.C.; visualization, M.-C.H. and Y.-Q.G.; supervision, Y.-Q.G.; project administration, Y.-Q.G., Y.-J.S., K.L. and F.-F.J.; funding acquisition, Y.-Q.G. All authors have read and agreed to the published version of the manuscript.

**Funding:** This work was supported by grants from the Natural Science Foundation of Fujian Province (No. 2022J01324) and the Shenzhen Mangrove Wetlands Conservation Foundation (S22251).

**Data Availability Statement:** Not applicable.

**Acknowledgments:** The authors would like to thank Y.-M.Y., Z.-X.H., G.-C.K., H.-W., T.-S.C. and S.-Z. for sample collection. We also thank the reviewers for their constructive criticism and improvement of the manuscript.

**Conflicts of Interest:** The authors declare no conflict of interest.

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
