# Peer review of "Comparison of the Meiofauna and Marine Nematode Communities before and after Removal of Spartina alterniflora in the Mangrove Wetland of Quanzhou Bay, Fujian Province"

_water, doi:10.3390/w15213829_

Round 1

Reviewer 1 Report

Comments and Suggestions for Authors

It is a descriptive research on the structure of the nematode community. I consider that the collection methods should be explained more widely, the results are clear, however there is not a good discussion, especially when the changes in the structure of the community are explained in a short time that lasts the evaluation after planting. of the mangrove.

Author Response

Ms. Ref. No.: water-2648598

Title: Comparison of the meiofauna and marine nematodes communities before and after removal of Spartina alterniflora in the mangrove wetland of Quanzhou Bay, Fujian province

Reviewer#1:

It is descriptive research on the structure of the nematode community. I consider that the collection methods should be explained more widely, the results are clear, however there is not a good discussion, especially when the changes in the structure of the community are explained in a short time that lasts the evaluation after planting of the mangrove.

We greatly appreciate your careful review of our manuscript and the valuable feedback you provided to help improve the paper. We have now made thorough revisions to this document based on your feedback. The details are as follows.

  1. Could you mention the mangal species used?

Answer: Good suggestion, the tree species names have been supplemented in the manuscript. The modifications are located on lines 13-14 of the manuscript.

  1. Could you describe the research hypothesis?

Answer: There is a competitive relationship in the spatial distribution of niche overlap between Spartina alterniflora and mangroves. It is assumed that mangroves can be used to occupy the living space of S. alterniflora and to improve the structure of mudflat ecosystem. The preliminary results of this study indicate that the biological alternative management method of first cutting and burying to remove S. alterniflora, and then planting local mangrove plants can improve the number of meiofauna in sedimentary environments and the biodiversity of marine nematodes. 

  1. The identification sites are too small, please increase the font.

Answer: Modified, the identification sites are amplified in the map. 

  1. The samples were collected in the beach? What distance from water line?

Answer: The sampling place is in the coastal mudflat, including the high tide area, the middle tide area and low tide area. The relevant instructions have been supplemented in the text. The modifications are located on lines 150-151 of the manuscript. 

  1. Please explain more about the sediment dynamics in the mangrove planting (seven months are sufficient for these changes)?

Answer: Thank you for the suggestions you provided to help us enhance the quality of our manuscript. We have made significant revisions in the discussion section. As you pointed out, sediment environmental factors in mangrove wetlands are crucial in influencing the structure and succession of benthic communities. In this study, while collecting samples of meiofauna, we simultaneously measured fundamental environmental factors such as sediment temperature, salinity of interstitial water, and sediment organic matter content. As per your question, the data obtained in just seven months of cultivating mangroves may not be sufficient to effectively explain the correlation between biotic factors of meiofauna (including nematode communities) and sediment environmental factors. Furthermore, due to the relatively short duration of the mangrove plantation in this experiment, it is currently not possible to discern differences in nematode communities among different mangrove habitat types.

Hence, this study focuses on the changes in composition and abundance of meiofauna before and after the removal of S. alterniflora, with a particular emphasis on studying the biodiversity and dominant genera of marine nematodes in two different habitats: S. alterniflora and mangroves.

  1. Could you mention some physical or biological factors afecting the nematode structure in mangrove systems? Organic matter content? sediment type? salinity?

Answer: You've raised an important question. Similar to the response in question 5, we measured environmental factors such as sediment temperature, dissolved oxygen in interstitial water, pH, salinity during our sampling. But at present, we cannot explain the relationship between the abundance of meiofauna, and the dominant genera of marine nematodes, and the measured environmental factors. Therefore, the focus of this paper is to compare the community structure of meiofauna and marine nematodes in S. alterniflora and mangrove during the initial stage of planting mangrove plants.

  1. Taxonomic groups?

Answer: We have made significant modifications in the conclusion section, located at lines 508-520 of the manuscript.

Reviewer 2 Report

Comments and Suggestions for Authors

The manuscript, which focuses on investigating how the removal of Spartina alterniflora, a common plant in salt marsh ecosystems, affects micro-scale communities in the mangrove wetland, is of interest for conservation purposes. While I find the manuscript acceptable, I have several suggestions for improvement. Here are my major comments below, along with minor suggestions and comments throughout the attached PDF of your manuscript.

Title: Please rephrase the title in a more engaging and less descriptive manner. I made a suggestion in the attached pdf.

Abstract:

lines 18-24: The text is overly detailed and specific; kindly rephrase it.

lines 21-24: Could you please specify whether all the listed species are specific to particular ecosystems, and whether they are considered autochtonous when the ecological balance is restored? Additionally, could you clarify why it's crucial to note that invertebrate species composition has changed? Are these species considered bioindicators? Keep in mind this comment along the text.

Final phrase: shortly describe the importance of the study and how results may impact the scientific community beyond this study.

Introduction:

-it needs more information or context about the current state of knowledge in the field (doi: 10.3390/plants12101923, https://doi.org/10.1016/j.apsoil.2022.104519, https://doi.org/10.1016/j.chemosphere.2017.06.060 ...)

Materials and methods:

lines 79-83: Explain briefly the rationale behind choosing the sites you sampled and why they are important for your study.

lines 114-124: It's surprising that no environmental parameters were considered for data interpretation. This is crucial, especially when addressing restoration and ecological changes. Plase make use of any available environmental parameters, including salinity, tides, substrate, pH, and temperature, and apply appropriate statistical analyses to demonstrate relationships between biota and abiotic factors.

Results:

The results are overly descriptive and challenging to follow, please rewrite. Consider presenting abundance data using plots rather than tables.

Discussion:

You should discuss your results, including the limitations of your study (such as the short time span), and outline potential future developments to enhance the knowledge gained in this initial approach.

Please interpret the results within a broader context and aim to provide explanations for the observed trends.

I hope you find my suggestions helpful in enhancing the quality of your manuscript.

Comments on the Quality of English Language

The English in the manuscript needs thorough revision to enhance readability. I've made some suggestions, but please keep in mind that English is not my native language.

Author Response

Ms. Ref. No.: water-2648598

Title: Comparison of the meiofauna and marine nematodes communities before and after removal of Spartina alterniflora in the mangrove wetland of Quanzhou Bay, Fujian province

Reviewer#2:

The manuscript, which focuses on investigating how the removal of Spartina alterniflora, a common plant in salt marsh ecosystems, affects micro-scale communities in the mangrove wetland, is of interest for conservation purposes. While I find the manuscript acceptable, I have several suggestions for improvement. Here are my major comments below, along with minor suggestions and comments throughout the attached PDF of your manuscript.

Answer: We greatly appreciate your careful review of our manuscript and the valuable feedback you provided to help improve the paper. We have now made thorough revisions to this document based on your feedback. The details are as follows. 

Title

  1. Please rephrase the title in a more engaging and less descriptive manner. I made a suggestion in the attached pdf.

Answer: Your suggestion is very good. But considering that the current research results are only preliminary, it is possible to maintain the original title to better represent the actual research content. After 4 years of project execution, when there is a large amount of complete biological and environmental data to write a paper, we will adopt the title you suggested. 

Abstract

  1. Lines 18-24: The text is overly detailed and specific; kindly rephrase it.

Answer: We have rewritten it and reorganized the comprehensive expression of the research results. The modifications are located on lines 20-33 of the manuscript. 

  1. Lines 21-24: Could you please specify whether all the listed species are specific to particular ecosystems, and whether they are considered autochtonous when the ecological balance is restored? Additionally, could you clarify why it's crucial to note that invertebrate species composition has changed? Are these species considered bioindicators? Keep in mind this comment along the text.

Answer:Marine invertebrates, especially meiofauna, are the most dominant metazoans in most marine ecosystems. In the muddy sediments of mangrove wetlands, marine nematodes make up more than 85% of the total meiofauna. Marine nematodes are widely distributed and abundant in quantity, with rich species diversity. Sometimes, research on meiofauna specifically pertains to the study of marine nematodes. Marine nematodes can serve as environment indicator, referencing articles 17-18.

In the manuscript, we consulted and supplemented the literature, referencing articles 23-25,27-29. These references were used to illustrate that the dominant genera of marine nematodes found in both habitats are widely distributed in similar environments in China or globally. It can also be stated that they are autochthonous when ecological balance is restored. 

  1. Final phrase: shortly describe the importance of the study and how results may impact the scientific community beyond this study.

Answer: Thank you for your suggestion. The revised manuscript has added the importance and scientific significance of this study. The modifications are located on lines 111-134 of the manuscript. 

Introduction

  1. It needs more information or context about the current state of knowledge in the field (doi: 10.3390/plants12101923, https://doi.org/10.1016/j.apsoil.2022.104519, https://doi.org/10.1016/j.chemosphere.2017.06.060 ...)

Answer: Thank you for providing the materials. We have added literature to supplement the background knowledge. 

Materials and methods

  1. Lines 79-83: Explain briefly the rationale behind choosing the sites you sampled and why they are important for your study.

Answer: The Quanzhou Bay Estuary Wetland Provincial Nature Reserve, where this study was conducted, is the experimental site for China's special management action to remove S. alterniflora. Therefore, this research area was selected for the study. If you would like more specific information, please refer to the notice issued by the State Forestry and Grassland Administration, Ministry of Natural Resources, Ministry of Ecology and Environment, Ministry of Water Resources, and Ministry of Agriculture and Rural Affairs of China on the "Special Action Plan for the Control of S. alterniflora (2022-2025)." 

  1. Lines 114-124: It's surprising that no environmental parameters were considered for data interpretation. This is crucial, especially when addressing restoration and ecological changes. Please make use of any available environmental parameters, including salinity, tides, substrate, pH, and temperature, and apply appropriate statistical analyses to demonstrate relationships between biota and abiotic factors.

Answer: Thank you for raising this question. As you pointed out, sediment environmental factors in mangrove wetlands are crucial in influencing the structure and succession of meiofauna. In this study we simultaneously measured fundamental environmental factors such as sediment temperature, salinity of interstitial water, and sediment organic matter content while collecting samples of meiofauna. However, due to the early stage of mangrove plantation during the study period, we have not yet established the relationship between environmental factors and meiofauna. This study focuses on the changes in composition and abundance of meiofauna before and after the removal of S. alterniflora, with a particular emphasis on studying the biodiversity and dominant genera of marine nematodes in two different habitats: S. alterniflora and mangroves. 

Results

  1. The results are overly descriptive and challenging to follow, please rewrite. Consider presenting abundance data using plots rather than tables.

Answer: Thank you for raising this question. The table presents in detail the differences in abundance and biomass of meiofauna and marine nematode communities in different habitats of S. alterniflora and mangroves. We also attempted to use graphics to present such results, but the abundance and biomass data between different groups at different times varied significantly, resulting in poor presentation results. We add separator lines in the abundance and biomass tables to better display data from different sampling times. 

Discussion

  1. You should discuss your results, including the limitations of your study (such as the short time span), and outline potential future developments to enhance the knowledge gained in this initial approach.

Answer: The revised manuscript has added a lot of content to the discussion section, including the inability to discern the differences in marine nematode communities among different mangrove plants due to the short planting time. The preliminary results of this experiment indicate that using the cutting method to remove S. alterniflora and then planting mangrove plants as a biological alternative management method can increase the number of meiofauna in the sedimentary environment and the biodiversity of marine nematodes, which is expected to achieve the function of repairing the ecosystem of estuarine wetland protection areas in the future. 

Conclusion

  1. Please interpret the results within a broader context and aim to provide explanations for the observed trends.

Answer:  We have made comprehensive revisions to the results section, primarily focusing on the transition of dominant genera of marine nematodes in S. alterniflora and mangrove habitats. The experiment also reveals the presence of some common dominant genera across different environments. In the conclusion section, we have also discussed the limitations of this experiment. The modifications are located on lines 508-520 of the manuscript. 

Round 2

Reviewer 2 Report

Comments and Suggestions for Authors

Authors have done a great job in addressing, to the best of their ability/methodology, the raised points.

I still believe that the authors should provide more detailed information on the following:

1. Study area: Explain in one phrase why this area was chosen as an experimental site. The reader should find the reason in the text, without the need to consult national action plans.

2. Discussion:

- Organic pollution in sediment/environment: You did not assess it or explain how it was assessed based on your methodology and/or results. If no measurements were taken, I would suggest removing direct comments on the topic that imply you found a relationship between organic pollution and identified taxa. If you did measure organic content, please explicitly address it in the relevant sections.

- Conclusions: I understand that at this point, it's challenging to relate abiotic and biotic factors. However, given the discussions you provided, including these interactions would bring an essential point into the observed trends. That's why I suggest mentioning this briefly as a shortcoming or something to consider in the future for a more comprehensive view of this ecosystem's dynamics.

All the best in your future endeavours

Comments on the Quality of English Language

The manuscript would benefit from an extensive English editing. 

Author Response

Ms. Ref. No.:water-2648598

Title: Comparison of the meiofauna and marine nematodes communities before and after removal of Spartina alterniflora in the mangrove wetland of Quanzhou Bay, Fujian province

Reviewer#2:

Authors have done a great job in addressing, to the best of their ability/methodology, the raised points.

Answer: We greatly appreciate your careful review of our manuscript and the valuable feedback you provided to help improve the manuscript. We have now made thorough revisions to this document based on your feedback. The details are as follows.

Study area

  1. Explain in one phrase why this area was chosen as an experimental site. The reader should find the reason in the text, without the need to consult national action plans.

Answer: Thank you for your suggestions. Relevant information about the experimental locations has been added to the manuscript, specifically in lines 67-69 and 103-105 (The revised content is in lines 56-58 and 80-82 after tracking changes).

Discussion

  1. Organic pollution in sediment/environment: You did not assess it or explain how it was assessed based on your methodology and/or results. If no measurements were taken, I would suggest removing direct comments on the topic that imply you found a relationship between organic pollution and identified taxa. If you did measure organic content, please explicitly address it in the relevant sections.

Answer: Thank you for your analysis and suggestions. We have reworked the relevant sections, removing direct comments on this topic (including the related statements in the abstract). The revised paragraphs can be found in lines 470-478 of the manuscript (The revised content is in lines 365-373 after tracking changes).

Conclusion

  1. I understand that at this point, it's challenging to relate abiotic and biotic factors. However, given the discussions you provided, including these interactions would bring an essential point into the observed trends. That's why I suggest mentioning this briefly as a shortcoming or something to consider in the future for a more comprehensive view of this ecosystem's dynamics.

Answer: This is the excellent question, and we appreciate this suggestion. We have incorporated relevant statements in the conclusion section, with the added content located in lines 519-525 of the manuscript (The revised content is in lines 395-401 after tracking changes).

All the best in your future endeavours.
Comments on the Quality of English Language
The manuscript would benefit from an extensive English editing.

Answer: Thank you for your suggestions on the English language proficiency of this manuscript. We will seek and confirm professional English editing before publication.
